# Role of Defects of Carbon Nanomaterials in the Detection of Ovarian Cancer Cells in Label-Free Electrochemical Immunosensors

**DOI:** 10.3390/s23031131

**Published:** 2023-01-18

**Authors:** Nattharika Runprapan, Fu-Ming Wang, Alagar Ramar, Chiou-Chung Yuan

**Affiliations:** 1Graduate Institute of Applied Science and Technology, National Taiwan University of Science and Technology, Taipei 106, Taiwan; 2R&D Center for Membrane Technology, Chung Yuan Christian University, Taoyuan 320, Taiwan; 3Sustainable Energy Center, National Taiwan University of Science and Technology, Taipei 106, Taiwan; 4Department of Chemical Engineering, Chung Yuan Christian University, Taoyuan 320, Taiwan; 5Department of Obstetrics and Gynecology, Cheng Hsin General Hospital, Taipei 112, Taiwan

**Keywords:** CA125, in situ reduction, gold nanoparticles (AuNPs), label-free immunosensor, multi-wall carbon nanotubes (MWCNTs)

## Abstract

Developing label-free immunosensors to detect ovarian cancer (OC) by cancer antigen (CA125) is essential to improving diagnosis and protecting women from life-threatening diseases. Four types of carbon nanomaterials, such as multi-wall carbon nanotubes (MWCNTs), vapor-grown carbon fiber (VGCFs), graphite KS4, and carbon black super P (SP), have been treated with acids to prepare a carbon nanomaterial/gold (Au) nanocomposite. The AuNPs@carbon nanocomposite was electrochemically deposited on a glassy carbon electrode (GCE) to serve as a substrate to fabricate a label-free immunosensor for the detection of CA125. Among the four AuNPs@carbon composite, the AuNPs@MWCNTs-based sensor exhibited a high sensitivity of 0.001 µg/mL for the biomarker CA125 through the square wave voltammetry (SWV) technique. The high conductivity and surface area of MWCNTs supported the immobilization of AuNPs. Moreover, the carboxylic (COO-) functional groups in MWCNT improved to a higher quantity after the acid treatment, which served as an excellent support for the fabrication of electrochemical biosensors. The present method aims to explore an environmentally friendly synthesis of a layer-by-layer (LBL) assembly of AuNPs@carbon nanomaterials electrochemical immunoassay to CA125 in a clinical diagnosis at a low cost and proved feasible for point-of-care diagnosis.

## 1. Introduction

Ovarian cancer (OC) is the formation of abnormal cells and multiply to form a tumor in the ovaries. The cells proliferate, invading and destroying healthy bodily tissue [1,2]. OC ranks 5th in cancer deaths among women, accounting for more deaths than any other female reproductive system [3]. Improving the prevention and detection of cancer at an early stage with sufficient sensitivity and specificity is a research priority because a disease diagnosed at a local stage has a 5-year relative survival rate of 93% [4].

Cancer antigen 125 (CA125) is a biomolecule in serum associated with OC. It is considered normal to have a CA125 level between 0 and 35 U/mL, while levels over 35 U/mL are associated with cancer [5]. Generally, CA125 is detected by electrochemical immunoassay [6], ELISA [7], chemiluminescent immunoassay [8], fluorescences [9], electrochemiluminescences [10], and radioimmunoassay [11]. These techniques involve a more complicated assay process and generally require labeling the antibody (Ab) or antigen (Ag). Interestingly, the electrochemical label-free immunosensor detection mechanism involves direct Ab-Ag interaction, which is a control process, low cost, simple apparatus, and fast detection [12]. Electrochemical immunoassay could be selected as an ideal strategy among various measurement techniques for tumor determination because of its portability, low cost, and high sensitivity [13]. Furthermore, electrochemical detection is faster and simpler to perform than many other current detection methods, which are time-consuming and labor-intensive [14]. Electrochemical detection is user-friendly, requires a minimal amount of samples, and can be adapted for use in the field.

Recently advances in nanotechnology have led to the developing of biosensor platforms with improved performance. The advances in synthesizing many new nanomaterials with controlled morphologies and unique physical properties have led to an explosion in biosensor application [15]. Carbon-based nanomaterials (CBNs) such as MWCNTs, VGCFs, graphite KS4, and SP have been widely utilized in developing electrochemical biosensor platforms. For example, a glucose biosensor fabricated using MWCNT-grafted chitosan (CS)-nanowire (NW)-glucose oxidase exhibited a high sensitivity of 5.03 μA/mM in a concentration range of 1–100 mM and a low response time to glucose measurement. The MWCNT-CS-NW facilitates the conduction of electrons between glucose oxidase and target molecules [16]. In another work by Eissa et al., electrochemical immunosensors detect survival motor neuron (SMN) protein using different carbon nanomaterial-modified electrodes. The comparative study of six different carbon nanomaterials such as carbon, graphene (G), graphene oxide (GO), single-wall carbon nanotube (SWCNT), MWCNT, and carbon nanofiber (CNF) was performed. Then, 4-carboxyphenyl layers were covalently grafted on the six electrodes by electro-reduction of the diazonium salt and continuously dropped, casting layer-by-layer. The performance of the six immunosensors suggests that carbon nanofiber is a better electrode material for the SMN immunosensor. The voltammetric SMN carbon nanofiber-based immunosensor showed high sensitivity (detection limit of 0.75 pg/mL) [17]. In 2021, Karuppiah et al. prepared a stable aqueous dispersion of VGCFs using PDA film synthesized through self-polymerization of dopamine monomers in an alkaline medium. The PDA-VGCF composite provides robust electrostatic interaction and active electrochemical sites for CAP antibiotic drugs displays a good linear response range (0.01–142 µM), lower detection limit (3 nM), and higher sensitivity (0.68 µAµM^−1^cm^−2^) with long-time stability up to 30 days [18]. In another study, PEDOT was synthesized by a novel approach that utilizes graphite nanosheets (GNS) dispersed in butanol. With its excellent conductivity and large surface area, GNS could be used as a material to construct enzymatic biosensing platforms because it could enhance enzyme immobilization in polymer matrices and their connectivity with substrates [19]. Developing wearable sensing for non-invasive bioelectrodes via flexible and stretchable materials was adapted with chitosan-carbon black (CH-CB) membranes. Their study presented a method of modifying membranes to produce biocompatible bioelectrodes suitable for use in enzymatic glucose biofuel cells (BFCs) and glucose detection systems using glucose oxidase and laccase enzymes [20] etc. Moreover, CBNs with oxygen-containing functional groups, specifically –COOH, allow for simple covalent bonding of biomolecules (e.g., antibodies), which is the most crucial process to help surface modification of the electrode. Further, carboxyl and hydroxyl groups have been generated on the surface of CBNs by the strong acid oxidation method in order to improve their dispersion in solutions [21,22,23,24].

Gold nanoparticles (AuNPs) have received more attention in electrochemical and biosensor applications because of the most stable metal nanoparticles and easy to synthesize in various sizes; there is a protocol to modify these nanoparticles with additional functionality, and they possess unique properties such as strong adsorption ability, large specific surface area, good biocompatibility, and conductivity [25,26,27,28]. Various techniques are available for synthesizing AuNPs, including chemical, thermal, electrochemical, and sonochemical methods [29]. The electrochemical method has been verified to be superior to other methods due to its modest equipment, low cost, lower processing temperature, high quality, and ease of controlling the yield [30,31]. For example, gold and graphene oxide via one-step electrodeposition is used to modify the electrode surface for the non-enzymatic determination of glucose. The reduction peaks occur at 1.1 and 0.9 V, confirming that gold nanomaterials undergo redox reaction on the electrode surface and that the deposition condition of 20 min is optimum [32]. In another study, the electrodeposited AuNP-film developed a label-free electrochemical immunosensor for strongyloidiasis. Layer-upon-layer attachment of strongyloides monoclonal recombinant antibody protein (rMAb23) onto AuNP-film was constructed, utilizing a thiol linker via a self-assembly monolayer (SAM) technique of 11 -mercaptoundecanoic acid (11-MUA). The CV was swept from −0.1 to 1.5 V vs. Ag/AgCl for 15 cycles at 100 mV s^−1^ [33]. MWCNT/GNPs nanocomposite electrode was produced by electrodeposition of GNPs onto the surface of MWCNT with a potential of −0.5 V for 20 s by chronoamperometry to perform selective and sensitive electrochemical detection of dengue toxin [34]. These modifications can substantially increase the immobilized amount of S-functionalized compounds and enhance the stability of the S-Au bond and self-assembled monolayer (SAMs) [35]. The attachment of metal nanoparticles to carbon materials is a way to obtain novel hybrid materials with interesting properties for various applications because CBNs are helpful as a support for gold nanoparticles in many potential applications, ranging from advanced catalytic systems to sensitive electrochemical sensors and biosensors [36].

This study aims to establish an effective method with a low-cost, easy-to-prepare, and easy-to-use biosensor as a label-free immunosensor for CA125 detection using one-step electrochemical deposition of gold (Au^3+^) with CBNs. We have modified and investigated different commercial CBNs such as MWCNTs, VGCFs, KS4, and SP to fabricate immunosensor (e.g., AuNPs@MWCNTs-nafion, AuNPs@VGCFs-nafion, AuNPs@KS4-nafion, and AuNPs@SP-nafion) on the GCE surface. The CBNs provided a larger electrode surface and the presence of more active sites, while electron transfer was accelerated by incorporating AuNPs. Nafion was utilized as a cation exchange polymer in which CBNs and AuNPs could be tightly attached to the electrode surface. The resulting modification based on AuNPs@MWCNTs electrode showed an excellent dynamic linear range of 0.001 to 10 μg/mL toward CA125 detection. The proposed immunosensor showed the potential application in the clinical screening of CA125 biomarkers at a low cost, which proved feasible for point-of-care diagnostic.

## 2. Experimental Section

### 2.1. Materials

All carbon-based nanomaterials (MWCNTs, VGCFs, KS4, and SP) were purchased from Ubiq Technology Co., Ltd., (Taoyuan, Taiwan). Sulfuric acid 95–98% was obtained from Scharlau (Taipei, Taiwan). Nitric acid 65% was purchased from Carlo Erba, (Cornaredo, Italia). Nafion (perfluorinated resin) solution (5 wt %), and 11-Mercauptoundecanoic acid 95% (11-MUA) were obtained from Sigma Aldrich (St. Louis, MO, USA). 99.7% Ethanol (EtOH) was obtained from ECHO Chemical Co, Ltd., (Miaoli, Taiwan). Tetrachloroauric (III) acid trihydrate (HAuCl₄·3H₂O), N-Hydroxysuccinimide (NHS), Potassium ferricyanide [K₃Fe(CN₆)] were purchased from Acros Organics, UK. Potassium hexacyanoferrate (ii) trihydrate [K₄Fe(CN₆)·3H_2_O], 1-(3-Dimethylaminopropyl)-3 ethyl carbodiimide hydrochloride (EDC) were purchased from Alfa Aesar, (Tewksbury, MA, USA). Potassium Chloride (KCl) was obtained from Fisher Chemical, (Waltham, MA, USA). PBS (10 X), pH 7.4, Biomate^TM^, and BSA (Bovine Serum Albumin) were purchased from Rainbow Biotechnologies Co., Ltd., (Taipei, Taiwan). Ab (X306) was purchased from HyTest Ltd., (Turku, Finland), and Ag (CA125) was purchased from purchased from Bio-techne, (Minneapolis, MN, USA). Deionized water (DI water) purified by a Millipore purification system (ICW-3000™ Water Purification System, Merck, Darmstadt, Germany) with a resistivity of 18.2 ΜΩ•cm was used in all syntheses.

### 2.2. Characterization

All electrochemical measurements were performed using a multichannel potentiostat (VMP3, Bio-Logic Science Instruments, Bio-Logic, Seyssinet-Pariset, France). A glassy carbon electrode (GCE) was used as the working electrode, while Ag/AgCl and a platinum wire served as reference and counter electrodes. A pH meter (PL-700, Taipei, Taiwan) was employed to analyze the immobilized protein of the electrochemical performance of the CA125 immunosensor. All samples were deposited onto the screen-printed electrode (SPE) with the composite procedure for morphology characterization, respectively. Raman spectroscopy studies on the graphite-based materials were detected by JASCO 5100 spectrometer (JASCO, Tokyo, Japan) in domain wave numbers 650 to 3000 cm^−1^. The chemical analysis is characterized using X-ray photoelectron spectroscopy (XPS, PHI 5000 Versaprob III, Kanagawa, Japan) and Fourier-transform infrared spectroscopy (FT/IR-6700, JASCO, Tokyo, Japan) ATR-FTIR in the frequency range of 4000–500 cm^−1^. X-ray diffraction spectroscopy (XRD) data were obtained using an X-ray diffractometer (D2 Phaser, Bruker, Karlsruhe, Germany) with a range from 10° to 80°, increment of 0.5°, and scanning rate of 5°/min. The morphology of the samples was investigated through Schottky field emission scanning electron microscope (FESEM, JSM-7900F, JEOL, Tokyo, Japan).

### 2.3. Oxidation of MWCNTs, VGCFs, KS, and SP

20 mg of carbon nanomaterial was suspended in a 20 mL concentrated mixture of H₂SO₄ and HNO₃ in a 1:3 ratio (*v*/*v*) by refluxing for 16 h at 80 °C to obtain a homogeneously distributed black solution. Then, the carbon nanomaterials were washed with DI water to remove acid until the pH was natural. Furthermore, it dried for 2 days to obtain carboxylated carbon materials [37].

### 2.4. Preparation of AuNPs-Carbon-Nafion (AuNPs@carbon)

First, the GCE working electrode was polished with 0.05 μm (polished alumina pad), and a 0.1 μm (shiny diamond pad) was thoroughly washed with EtOH, DI water, and air-dried. The preparation of AuNPs@carbon was followed by literature [12]. 5 mg carbon powder were dispersed into 4 mL of DI water, containing 1 mL of 0.2 wt% Nafion (ratios 4:1) and sonication for 2 h from homogeneous suspensions. Then, 3 mM HAuCl_4_ solution (containing 0.1 M H₂SO₄) was added into a mixture in 1:2 ratios and continuously sonicated for 2 h. After that, the preparation of the precursor solution was completed. The one step AuNPs fabrication of carbon nanomaterials such as the AuNPs@MWCNTs-nafion (AuNPs@MWCNTs), the AuNPs@VGCFs-nafion (AuNPs@VGCFs), the AuNPs@KS4-nafion (AuNPs@KS4), and the AuNPs@SP-nafion (AuNPs@SP) is prepared by CV. The scan was started from 1.0 to −0.2 V at 100 mV s^−1^ for 50 cycles.

### 2.5. Electrodeposition of AuNPs@Carbon Composites on GCE Electrode

1.7 mL of AuNPs@carbon solution was electrochemically reduced on the GCE electrode by CV with an Ag/AgCl electrode and a platinum wire acting as reference and counter electrodes, respectively. The scan was started from 1.0 to −0.2 V at 100 mV s^−1^ for 50 cycles. The deposition of AuNPs@carbon/GCE is shown in Appendix A. At first cycle, a reduction peak is observed at ~0.27 V, and the next cycle involves a shift to a more positive potential, enabling easier gold electrodeposition on the existing gold particles [38]. The reduction of Au is shown in Equation (1) [39].
(1)$$AuCl_{4}-\left(aq\right)+3e^{-}\to Au\left(s\right)+4Cl\;$$

In order to, the electrochemical deposition of AuNPs@carbon composite in various cycles and their CV examined by 0.1 M KCl/5 mM [Fe(CN)]^3−/4−^(0.1X PBS) was shown in Appendix A. The redox peak current densities are remarkably enhanced, accompanied by almost unchanged redox potentials. The results of electrochemical deposition of HAuCl₄ at various different cycles indicated that the deposition for 50 cycles had a slightly higher peak current than the others (10 and 30 cycles). Therefore, it is decided to use the electrochemical immunosensor with 50 cycles. It is understood that the higher deposition of noble metals directly proportional to the higher number of functionalized 11-MUA could be possible to have greater functionalizing to Ab [40].

### 2.6. Fabrication of the Immunosensor

After the AuNPs@carbon/GCE electrode is completed. The GCE electrode is immobilized in 10 mM 11-MUA solution (prepared in EtOH) at 4 °C for 12 h, then rinsed with 0.1 X PBS for removed physically adsorbed 11-MUA. This step gives an 11-MUA self-assembly monolayer film with AuNPs on the GCE surface. Next, the electrode was immersed into 0.1 M EDC for 5 h at 25 °C and then 0.4 M NHS for another 5 h at 25 °C. In this reaction, the EDC reacts with the carboxylic acid and forms o-acylisourea, a highly reactive chemical that reacts with NHS and forms an NHS ester, which quickly reacts with an amine (i.e., Ab) to create an amide [41]. The GCE electrode were incubated with antibody (X306) overnight at 4 °C to form a covalent linkage. After that, GCE electrode was rinsed with 0.1 X PBS. The GCE electrode is incubated with 1% BSA for 40 min at 4 °C to block some nonspecific active sites. In addition, the modified GCE was washed with 0.1 X PBS to eliminate unreacted BSA. Finally, the GCE electrode is incubated with different concentration of CA125 solutions for 45 min at 30 °C, and afterward, rinsed with 0.1 X PBS to remove unreacted CA125 for further electrochemical measurements (Figure 1).

## 3. Results and Discussion

### 3.1. Scanning Electron Microscopy Characterization of the Carbon Nanomaterial-Modified Electrodes

SEM imaging was used to analyze the morphology of the electrode surface of various materials. Figure 1 displays SEM images of four carbon nanomaterials from pristine oxidation with strong acid and AuNPs@carbon composite. The preparation of all nanocomposites was electrodeposited onto the surface of SPE electrode for measurement. In Figure 1a, the morphology of the SPE exhibits an intrinsically porous microstructure. Similarly, Figure 1b shows the 50 AuNPs cycles and Au^3+^ electro-reduction on the substrate electrode, in which the inserted dimension is approximately 300 nm in diameter of AuNPs. Figure 1c displays the characteristics of pristine MWNCTs: long, smooth surfaces attached in a bundle with an average diameter of 25.37 nm. However, after being oxidized with strong acid, the nanotubes appeared curled, tangled, and smaller in diameter, 18.13 nm, and displayed in Figure 1d. As shown in Figure 1e, structured film with 50AuNPs@MWCNTs/SPE. The MWCNTs and AuNPs were embedded and well dispersed. The nanocomposites were firmly and uniformly adsorbed through interactions, and the Au nanoparticles were scattered on the surface. Furthermore, this uniform nanostructure provided an efficient surface for loading AuNPs-Ab conjugate and accelerating electron transfer. The surface morphology of unmodified VGCFs in Figure 1f reveals the smooth nanofibers structure in a large diameter of 171.92 nm. After the acid-treatment (Figure 1g), the VGCFs shape was changed by sticking spherical, reducing a smaller diameter to 52.83 nm, and acicular particles. Figure 1h displays 50AuNPs@VGCFs/SPE, which indicates that the deposition of 50AuNPs@VCGFs composite is not extensive as 50AuNPs@MWCNTs. The results showed that AuNPs were dispersed over the VGCFs and exhibited agglomeration somewhere, meaning they were not homogeneously attached. Figure 1i,j shows KS4 before and after the acid treatment. Before the treatment, KS4 has a 3D layer of graphite and fold structure with a long smooth sheet and flat. After it was treated, the KS4 material showed a slight sheet occurrence. After in-situ electro-reduction, 50AuNPs@KS4/SPE (Figure 1k) in which the Au nanoparticles uniformly distributed through KS4. Figure 1l shows the SEM images of SP before the acid treatment, indicating a uniform particle-size distribution of them is similar, with an approximate diameter of 55.34 nm. The acid treatment (Figure 1n) does not affect the particle size, the diameter of the SP was slightly less, i.e., 47.37 nm, and the connecting structure of carbon nanoparticles is well-preserved. Figure 1o shows 50AuNPs@SP/SPE. This observation indicates that the walls and area of these CBNs are not impacted after being treated with strong acid. In addition, it is clear that the spherical morphology of AuNPs on the layered thin film of each CBNs.

### 3.2. Characterization of the Electrode Using FTIR, Raman Spectroscopy, XRD, and XPS

The ATR-FTIR spectra of pristine, chemically treated carbon samples, and electrochemically deposited gold were measured to confirm the functionalization and interaction between the carbon and AuNPs. Figure 2a, the FTIR absorption spectra of pristine and oxidized MWCNTs, VGCFs, KS4, and SP, respectively, show bands at 1728, 1667, 1662, and 1688 cm^−1^. Those stretching bands arose from -C=O stretching, indicating the presence of carboxylic acid on the surface of CBNs after the acid treatment [37]. In the 50AuNPs@carbon composite, nafion characteristic vibration peaks at 3251 cm^−1^ were the intensity of -O-H stretching vibration, which corresponds to physically adsorbed water [42]. The peak at 1630 cm^−1^ as -O-H bending vibration of free water molecules, whereas the band at 1014 cm^−1^ represents symmetric -S-O stretching [43]. The composites have a comparable peak at 1014 cm^−1^, and bands at 1211 and 1151 cm^−1^ were formed due to symmetric C-F stretching [44]. Furthermore, the -C-O-C stretching caused the band at 866 cm^−1^ and the band at 567 cm^−1^ corresponds to symmetric O-S-O bending, whereas the 696 cm^−1^ band was due to -C-S group stretching [45]. FTIR spectral data confirms that acid treatment effectively produced functional groups and the 50AuNPs@carbon composite have successfully combined the carboxyl group, gold, and nafion onto the surface of all the CBNs.

The graphitic degree of all the CBNs samples were estimated by the intensity ratio of ID/IG from the Raman spectra displayed in Figure 2b. The results show two common bands that are characteristic to the G band (graphitic band) at 1580 cm^−1^ accounted for the degree of crystallinity of graphitic sheets, and the D-band (disorder band) at 1340 cm^−1^ represented the crystal defect of the graphitic sheet [46]. The intensity ratio of the D band and G band (ID/IG) was employed to determine the degree of structural defects or disorders in CBNs. The position of the D and G (ID/IG) ratio is presented in Appendix A. After chemical treatment, ID/IG increased due to the oxidation inserting chemical groups that can be interpreted as defects in the carbon nanostructure [47]. These characteristic peaks of CBNs can still be observed after oxidation treatment, proving that the treatment does not damage the graphitic structure. 50AuNPs@carbon composite samples were deposited onto the SPE for Raman spectra measurement and the extra peak at 1036 cm^−1^ could be accounted to SO₃− of nafion characteristics [48]. The bare SPE containing graphite represented the typical Raman spectrum, with the peak width being a higher wave number. The results showed that the ID/IG ratio slightly increased from bare to composite, indicating carbon nanomaterials and AuNPs distribution on the SPE surface. Based on results from this study, four CBNs were compared for their effect on introducing defects on their surfaces following the acid treatment. Raman spectra indicate that pristine MWCNTs and SP have a high ratio of ID/IG, which could make it easier for acids to react with and have a higher defect after the oxidization. In contrast, VGCFs and KS4 show minor defects at the pristine state. The acid treatment has not improved the defects to that of VGCFs and KS4. Because defects play a vital role in catalytic activities, it was apparent that higher defects in MWCNT and SP may allow better sensing abilities.

XRD was used to investigate the phase identity and crystallinity of pristine and acid-functionalized carbon nanomaterials. As shown in Appendix A, the diffraction peaks of samples at around 25.8°, 42.8°, 53.8°, and 78.4° were identified as the (002), (100), (004), and (110) d-spacing planes of the hexagonal graphite, respectively. Compared with the XRD pattern of unmodified carbon nanomaterials, the main peak of oxidized carbon materials 2θ = ~25° becomes sharper and lower than these of pristine carbon nanomaterials, which indicates that due to the creation of an active site on the surface of carbon nanomaterials, by chemical treatment. The results show that the oxidized MWCNTs and oxidized SP present poor crystallinity. While oxidized VGCFs and KS4 revealed similar XRD patterns; thus, the oxidation by acids did not affect the crystalline structure. The particle size of all the samples were calculated according to Scherrer Equation and can be seen in Appendix A. In this perspective, The CBNs can be arranged MWCNTs > SP > KS4 > VGCFs in this order based on the defects calculated from the Raman spectra. However, XRD spectra results indicate that crystallinity of MWCNT and SP were affected significantly than that of the KS4 and VGCF.

Figure 3 presents the XPS C 1s spectra of CBNs pristine, oxidized, and 50AuNPs@carbon composite. A characteristic peak for the C 1s located at the binding energies of 284.7 and ~285 eV can be observed in the pristine, oxidized, and composite spectra, respectively. They correspond to C-C bonds with sp^2^ hybridization and C-C bonds with sp^3^ hybridization. Both C-O and C-OH functionalities were associated to the peak at 286.2 eV. The carboxylic acid functional group was identified through the peak at ~289 eV. A peak at ~291 eV was the aromatic structure of π-π shake-up [49]. Moreover, the peak at 283.7 eV corresponds to C-Si bonds associated with the carbon and SiC on the surface. Acid treatment improved the relative amounts of COOH from pristine to oxidized in every CBNs; MWCNTs, from 4.89% to 7.1%, VGCFs from 2.93% to 5.1%, KS4 from 2.88% to 4.38%, and SP from 3% to 5.47%. Hence, it appears to be a decrease in the relative content of the C-C after the acid treatment. The formation of AuNPs@carbon composite displayed in Figure 3c,f,i,l) of 50AuNPs@MWCNTs, 50AuNPs@VGCFs, 50AuNPs@KS4, and 50AuNPs@SP) show the C 1s peaks spectrum, with the peaks approximately at 284.8, 286.1, 289, 291.9 and 292.8 eV are representing the C-C, C-O, COOH, CF₂, CF-O, and CF₃ [50] respectively. The peak at 291.9 and 292.8 eV assignments correspond to the chemical structure of the nafion chains. XPS spectra show that Au 4f _7/2_ and Au 4f _5/2_ appear at 84 and 87.7 eV [51]. in Appendix A. The formation of a 50AuNPs@carbon composite further confirms the existence of nafion and AuNPs on the surface of all CBNs.

### 3.3. Electrochemical Characterization of the Fabricated Immunosensor

To obtain the electrochemical active surface area of bare GCE and AuNPs@carbon modified electrode in 5 mM [Fe(CN)_6_]^3−^/0.1 M KCl containing 0.1 X PBS at various scan rates from 10 to 100 mV s^−1^ were used and shown in Figure 4a,c,e,g,i. The redox peak current densities were gradually increased as the scan rate (*v*) changed from 10 to 100 mVs^−1^. Figure 4b,d,f,h,j showed the linear plot of the redox peak current density versus the square root of scan rate, indicating that the AuNPs@carbon modified GCE was a diffusion-controlled redox process. Furthermore, the surface area was calculated using the Randles-Sevcik Equation (2) [51,52,53].
𝑖_c_ = 2.69 × 10^5^ AD^1/2^*n* ^3/2^C*v*^1/2^
(2)
where 𝑖_c_ = the peak current density (A), A = the area of electrode in cm^2^ (active surface area), D = the diffusion coefficient (cm^2^ s^−1^), *n* = the number of electrons transfer in the oxidation/reduction process (1e^−^), C = electrolyte concentration (mol/cm^3^), and ν is the scan rate (V s^−1^). For [Fe(CN)_6_]^x−^, D = 7.6 × 10^−6^ cm^2^ s^−1^, and C = 5 × 10^−6^ mol/cm^3^. Based on the calculation, Table 1. shows the electrochemical surface area of four different carbon nanomaterials. The active surface area was estimated to be 0.953 cm^2^ of 50AuNPs@MWCNTs, 0.865 cm^2^ of 50AuNPs@VGCFs, 0.838 cm^2^ of 50AuNPs@KS4, and 0.849 cm^2^ of 50AuNPs@SP that were about 1.79, 1.62, 1.59, and 1.58 time higher than bare GCE value of 0.532 cm^2^. The AuNPs@carbon composite modified electrode with a higher surface area, could provide high electrocatalytic performance in detecting CA125. The 50AuNPs@MWCNTs has relatively higher active surface area and produce high detection sensitivity, respectively.

### 3.4. Electrochemical Characterization of the Fabricated Immunosensor

The AuNPs@carbon nanocomposite electrodes were evaluated by a standard redox couple, ferro/ferricyanide. Figure 5a shows the CVs of 50AuNPs@MWCNTs composite and further fabricated processes in 0.1 X PBS in 5 mM [Fe(CN6)]^3−^/^4−^/0.1 M KCl solution. The CV and EIS spectra of other nanocomposites such as VGCFs, KS4, and SP based electrodes were shown in Appendix A. The current density of the redox peaks decreased after attaching Ab and BSA over 50AuNPs@MWCNTs composite, denoted as BSA/Ab/AuNPs@cMWCNTs/GCE, respectively. As expected, the current density signal was further decreased after immobilization of CA125 (Ag/BSA/Ab/AuNPs@MWCNTs/GCE), suggesting the successful formation of the fabricated immunocomplex. Furthermore, electrochemical impedance spectroscopy (EIS) was used as a monitor to investigate the effects of interfacial state changes on each step of the constructed immunosensor procedure. Figure 5b, the semicircle portion at medium frequency reflects the electron-transfer resistance (Rct). The bare GCE showed a Rct value of 221 Ω. The 50AuNPs@MWCNT modified GCE, the Rct value decreased to 155 Ω which demonstrates that the AuNPs and carbon effectively placed on the GCE surface. The immobilization of Ab and BSA and increased Rct (i.e., the Rct value up to 272 Ω, 428 Ω) indicate the effective immobilization of Ab and BSA, respectively. As expected, the diameter of the semicircle increased and Rct value calculated to be 702 Ω for Ag/BSA/Ab/50AuNP@MWCNTs. The similar trend was observed to other carbon electrodes. This could be understood that the insulated immunocomplex formed as the blocking layer preventing the diffusion of the electron pathway. These changes confirmed that the Ab-Ag immunocomplex was successfully formed. Further, the resistance of fabrication step networks layer-by-layer construction of immunosensor was presented in Appendix A. The lower Rct values (100–160 Ω) of these 50AuNPs@carbon composites revealing a rapid electron transfer kinetics at the electrode/electrolyte interface and suggesting the suitability of the 50AuNPs@carbon matrix for immunosensor fabrication. The enhanced electron transfer kinetics for 50AuNPs@carbon composite electrode is attributed to the presence of AuNPs on carbon, which acts as small conduction pockets of charge from the solution to an electrode surface [27]. Next, Ab, BSA, and Ag were immobilized, and the Rct was increased step-by-step. The resistance in BSA immobilization step shows that the Rct value is increased in the following order: MWCNTs ˂ SP ˂ KS4 ˂ VGCFs. The reason could be more active sites (carboxylic groups) in MWCNTs after acid treatment could enable biomolecules to adhere to the network of MWCNTs in the composite and to work more efficiently in sensing applications.

### 3.5. Optimization of Operational Parameters for Immunosensor

The proposed immunosensor was optimized by parameters such as pH, antibody concentration, incubation time of BSA, incubation temperature, and Ab-Ag incubation time. The pH significantly impacted the biosensor because it can influence immobilized antibodies’ electrochemical behavior and stability. The pH values on the immunosensor were determined by measuring the current signal at different values ranging from 5.5 to 8 in Figure 6a. The peak current density value was high at pH 5.5 and slightly decreased with an increase in the pH values. These results are demonstrating that the Ab undergoes denaturation in alkaline solution, thus pH of 5.5 was chosen for this experiment [52]. The maximum concentration of Ab for immobilization was optimized by measuring a range of 5 µg/mL to 25 µg/mL. The results show that the successful attachment of the antibody reduces the electron transfer from the redox solution to the electrode surface. The current was constant with the further increasing concentration of antibody as seen in Figure 6b and hence 20 µg/mL was fixed as the optimum antibody concentration. Figure 6c shows the SWV current density changes of the immunosensor with 1% BSA concentration with different incubation times from 10 to 60 min. The current decreased initially, reaching the lowest current limit at 40 min. The current response was approximately constant upon further incubation time and 40 min was optimized as incubation time. To optimize the immobilization temperature of Ab and Ag, we detected the SWV signal from 5 to 40 °C displayed in Figure 6d. The maximum 𝛥I signal response was obtained at 30 °C. Therefore, 30 °C was selected as the optimal Ab-Ag incubation temperature and was maintained at 30 ± 2.0 °C. Further, incubation time for the formation of antigen and antibody immuno-complex was performed which concludes in 45 min. This result indicates that 45 min was the optimum incubation time for Ag CA125, showed in Figure 6e.

### 3.6. Electrochemical Immunosensing of the CA125 Antigen

The analytical performance of the CA125 immunosensor was examined under the optimum condition (pH: 5.5, antibody concentration: 20 μg/mL, BSA incubation time: 40 min, incubation temperature: 30 °C, and antigen-binding time: 45 min). Figure 7a,c,e,h displays the SWV curves and the peak current response of the proposed immunosensors after the Ag-Ab immune reaction of MWCNTs, VGCFs, KS4, and SP. It shows the concentration range from 0.001 to 10 μg/mL (0.0001 to 10 μg/mL of VGCFs). The peak current decreased with the increasing concentration of CA125, which clearly shows a signal off-trend. These results prove that CA125 successfully binds with Ab (X306) on the surface of the electrode. In Figure 7b,d,f,i the linear regression equation was Ipc (μA) = 0.6277x (Log C_CA125_) − 9.6231 (R^2^ = 0.9624) of MWCNTs. The fabricated immunosensor based on AuNPs@MWCNTs show a good linear range. This might be due to CBNs with different relative contents of the oxygen-containing group’s effect on the electrochemical performance of the immunosensor. The carboxylic group on the MWCNTs reaches the highest point of 7.1%. These properties make MWCNTs as suitable material support for biomolecule immobilization. Furthermore, this uniform nanostructure provided an efficient surface for forming AuNPs-Ab conjugate and accelerating electron transfer. The analytical parameter of the fabricated immunosensor has been compared with the other kinds of carbon nanomaterials for CA125 which was shown in Table 2.

Indeed, with respect to the morphologies and the electrocatalytic performance results, the obtained order for the carbon nanomaterials is MWCNTs is the highest current signal compared with the other carbon nanomaterials such as VGCFs, KS4, and SP. The observations lead to the following conclusions: (i) oxygen functional group in the CBNs, which improve immobilization; (ii) the ID/IG ratio calculated from the Raman spectra revealed that VGCFs and KS4 have relatively lower defects for electrocatalytic reaction compare to MWCNTs and SP. Based on the present study, it is clear that MWCNT is highly sensitive for CA125 detection, which differs from Eissa et al. ‘s study demonstrating functionalized CNF was most responsive for SMN detection. In both cases, the functionalization of carbon nanomaterials played an important role irrespective of their structure and the sensitivity primarily associated with the amount of defects in the nanomaterials. The lower defects represent high crystallinity which is not suitable for binding with biomolecules and concurrently the sensor characteristics of the carbon nanomaterial follow the defects present in it. Also, the SEM result showed that the VGCFs exhibited agglomeration somewhere, indicating that they were not fully attached to the surface electrode; (iii) The presence of sp^2^ hybridized carbon atom edge planes and oxygenated groups over the MWCNTs makes them capable of attaching biomolecules on their surface to act as electrochemical biosensors and can improve further [54].

Table 3 summarizes some characteristics of the fabricated electrode and the other fabrication sensor, including linear range, LOD, and different detection methods. Most importantly, our proposed sensor reduces AuNPs and is more stable and time-consuming than other techniques (such as drop-casting and drying). As a result, the proposed AuNPs@MWCNs/GCE-based electrochemical determination of the CA125 biomarker is a facile construct, rapid detection, and without secondary Ab and nanomaterials for signal amplification. The sensitivity of currently proposed 50AuNPs@MWCNTs sensor is 0.0004 U/mL which is significantly higher than that of a commercial biosensor (Human Mucin-16 ELISA Kit, Thermo Fisher, Massachusetts, USA) used for the detection of CA125 whose sensitivity values is about 1.88 U/mL [55]. We suggest that the electrochemical deposition method allows better control of Au to enhance the electrocatalytic performance of AuNPs@carbon nanomaterials because of the high electrical conductivity and large surface area, which might allow the rapid heterogeneous electron transfer.

## 4. Conclusions

In summary, we have studied different types of CBNs, such as MWCNTs, VGCCFs, KS4, and SP to fabricate label-free immunosensors for CA125 detection for OC. Acid treatment of CBNs resulted in surface modification and end up with oxygen-containing functional groups which are essential for bonding with biomolecules. The electrochemical deposition method was utilized and optimized for 50 cycles to fabricate 50AuNPs@carbon nanocomposite electrode. The FTIR, Raman, XRD and XPS studies confirmed that sufficient functional groups were present in the CBNs and interaction with AuNPs. The SWV were measurements performed by the GCE/AuNPs@carbon/11-MUA/EDC/NHS/Ab/BSA/CA125 in the dynamic linear range from 0.001 to 10 μg/mL towards detecting CA125. Among the four different CBNs, the MWCNTs demonstrated a high sensitivity towards CA125 detection. The excellent surface modification, efficient electrodeposition of AuNPs@MWCNTs facilitates high binding with antibody which paves higher amount of sensitive detection. Therefore, the 50AuNPs@MWCNTs label-free immunosensor can be used as an ultrasensitive electrochemical for future clinical diagnostic.

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
