# Peer review of "Role of Defects of Carbon Nanomaterials in the Detection of Ovarian Cancer Cells in Label-Free Electrochemical Immunosensors"

_sensors, 2023, doi:10.3390/s23031131_

Round 1

Reviewer 1 Report

The work is scientifically sound, and while there is new information here, the methodology novelty is somewhat adequate.

In a present state the manuscript submitted is adequate and to be recommended for publication with major correction. 

Listed of the corrections and question:

1.     Title should be write in full sentences especially CA125.

2.     Introduction

i.                 Introduction too long and should be reduce.

ii.                Don’t repeat the full sentences if already used abbreviation. i.e. line 84.

3.     Mechanism of the reaction should be done/state.

4.     Validation with commercial/comment instrument/technique should be done for confirmation.

Author Response

Dear Editor

Thank you for your letter regarding the reviewers’ comments. These comments suggest modifications. In the attached “revision letter” copy, each change has been distinguished it clearly from the earlier version of our manuscript. In addition to the specific modifications to the text, general explanations in response to their comments are listed as follows:

Reviewer #1: The work is scientifically sound, and while there is new information here, the methodology novelty is somewhat adequate. In a present state the manuscript submitted is adequate and to be recommended for publication with major correction.

Listed of the corrections and question:

  1. Title should be write in full sentences especially CA125.

Ans: Thanks for reviewer’s comment. The title of the manuscript is changed and hence the abbreviation is not used.

  1. Introduction
  2. Introduction too long and should be reduce.

Ans: Thanks for reviewer’s comment. The introduction is shortened appropriately without considerable change in the readability.

  1. Don’t repeat the full sentences if already used abbreviation. i.e. line 84.

Ans: Thanks for reviewer’s comment. The full sentence “multiwalled carbon nanotube” is removed in the revised manuscript.

  1. Mechanism of the reaction should be done/state.

Ans: Thanks for reviewer’s comment. Mechanism of the immunosensor is given in the revised manuscript in page no. 5 in section 2.5. At first cycle, a reduction peak is observed at ~ 0.27 V, and the next cycle involves a shift to a more positive potential, enabling easier gold electrodeposition on the existing gold particles (Chiang et al. 2019). The reduction of Au is shown in equation 1 (Zakaria et al. 2021). 

AuClâ‚„(aq) + 3e     Au(s) + 4Cl−             (1)

4.Validation with commercial/comment instrument/technique should be done for confirmation.

Ans: Thanks for reviewer’s comment. The sensitivity of the currently proposed 50AuNPs@MWCNTs sensor is 0.0004 U/mL which is significantly higher than that of a commercial biosensor (Human Mucin-16 ELISA Kit) used for the detection of CA125 whose sensitivity value is about 1.88 U/Ml [Mucin-16]. This comparison is included in the revised manuscript in section 3.6 under the discussion of Table 3.

Reviewer #2: The manuscript entitled “Improving the oxygen-containing functional groups of multi-walled carbon nanotubes to enhance the sensitivity of label-free electrochemical immunosensor for the detection of CA125” discusses about four different types of CBN electrochemical behavior against CA125. In short, the manuscript would have been clean and neat if the authors just focus on MWCNT. The investigation of other CBN derivatives makes the work quite extensive. However, materials characterizations and various electrochemical analyses provides significant input to the existing body of knowledge. So I have given some recommendation to improvise the manuscript before it can be considered for publication.

Title:

  1. Based on the manuscript – the study investigates on various types of CBNs. Therefore, it’s inappropriate to state one of the CBN-MWCNT’s name only in the title. It’s recommended to use general term – CBN. Suggested title: ” Improving the oxygen-containing functional groups of carbon-based nanomaterials to enhance the sensitivity of label-free electrochemical immunosensor for the detection of CA125”

Ans: Thanks for reviewer’s comment. The title of the manuscript can be changed as follows: Role of defects in carbon nanomaterials in the detection of ovarian cancer cells in label-free electrochemical immunosensors

Abstract:

  1. Page 1, Line 19-22: The labelling of modified electrode is too long. Be sure to include the significant materials only as your WE. In addition, I don’t find much info about mentioned materials in the Introduction section.

Ans: Thanks for reviewer’s comment. The labelling of modified electrode is now shortened appropriately in the revised manuscript. The importance of 11-MUA is written in line 132 in the original manuscript. The role of EDC and NHS is explained on page 5 in section 2.6. In this reaction, the EDC reacts with the carboxylic acid and forms o-acylisourea, a highly reactive chemical that reacts with NHS and forms an NHS ester, which quickly reacts with an amine (i.e., Ab) to create an amide.

Introduction:

  1. Authors have used 4 different CBN – MWCNT, VGCFs, graphite KS4, and carbon black super P (SP). The Introduction section must contain related works to KS4 and SP. I can only find many literatures were based on MWCNT and only one work by Karuppiah et al on VGCFs. More LR must be added on KS4 and SP.

Ans: Thanks for reviewer’s comment. Literature about KS4 and SP is already in the original manuscript. The research article from Scotto et al. 2020 discusses about the graphite nanosheets (GNS) which is similar to KS4. It can be found in line no. 95 in the original manuscript. Buaki-Sogó et al. 2021, discusses about chitosan-carbon black composite for biosensors which is similar to SP. It can be found line no.99 in the original manuscript.

Methods:

  1. Page 4, Line 178-179. It was little confusing when authors mentioned about 3 electrode system then, all sampled were deposited on screen printed electrodes. The SPEs are for morphology purposes only. So I recommend to shift this method to the end of Section 2.2 where author explained about FESEM.

Ans: Thanks for reviewer’s comment. The electrochemical studies were carried out using three electrode system. However, all the characterization methods used the screen printed electrodes. So, the sentence is shifted to before Raman spectroscopy characterization.

  1. Page 5, Line 207: What is rxn?

Ans: Thanks for reviewer’s comment. The word rxn is changed to scan in the revised manuscript. A similar change has been made in line no. 212.

  1. Section 2.6 – many grammatical errors were spotted. RT is not a scientific style of writing. Use specific standard room temperature i.e. 25°C.

Ans: Thanks for reviewer’s comment. The abbreviation RT is changed to 25 ºC.

Results and Discussion:

  1. The resolution of inset of Fig 1b must be enhanced. Fig 1e does not reflect the MWCNT anymore. Kindly replace the Fig 1b where MWCNTs are visible within the scattered AuNPs.

Ans: Thanks for reviewer’s comment. The resolution of Fig. 1b is improved. The MWCNT is visible in the right bottom after having a brightness correction in Fig. 1e. 

  1. Page 7, Line 292 – authors stated the membranes. Why it was suddenly labelled as membranes?

Ans: Thanks for reviewer’s comment. The word membranes is changed to composites in the revised manuscript.

  1. Author declared that, KS4 showed minor defect. However, according to Table S1 – I believe ID/IG increased from 0.240 to 0.472 which is a great enhancement. Kindly clarify.

Ans: Thanks for reviewer’s comment. The present work describes that there is a relation between the defects of carbon nanomaterials and their sensitivity. Table S1 summarizes defects of each carbon nanomaterials before and after the functionalization. It is sure that the defects were greatly improved for KS4. However, the final defects value has importance in the sensing properties. From the table S1, the defects order follows MWCNT>SP>KS4>VGCFs which is directly related to sensitivity order.

  1. Page 9, Line 337: How do you arrange MWCNTs>SP>KS4>VGCFs in this order?

Ans: Thanks for reviewer’s comment. The carbon nanomaterials can be arranged MWCNTs>SP>KS4>VGCFs in this order based on the defects calculated from the Raman spectra. However, XRD spectra results indicate that crystallinity of MWCNT and SP were affected significantly than that of the KS4 and VGCF. The above sentences included in the revised manuscript in the same place.

  1. The resolution of Fig. 4 must be enhanced.

Ans: Thanks for reviewer’s comment. The resolution of Fig. 4 is improved.

  1. Section 3.5: First of all, authors mentioned about 7 parameters were optimized. However, the results of 5 parameters only shown in the manuscript. The parameters such as Ab incubation time, Ab-Ag incubation time, Ag concentration missing in Fig. 6.

Ans: Thanks for reviewer’s comment. It is a mistake that it has only five parameters been optimized. It is corrected in the revised manuscript. Hence, the unattended parameters such as Ab incubation time, Ab-Ag incubation time and Ag concentration not to be considered.

  1. Section 3.5, Line 451: “The maximum concentration for immobilization….” Since authors investigated many parameters, they must assure which parameter are they discussing when moving from one to another. Kindly rewrite the sentences to depict a clearer picture of these parameters.

Ans: Thanks for reviewer’s comment. The maximum concentration for immobilization discusses about antibody. The sentence is rewritten as “The maximum concentration of Ab for immobilization”

  1. Fig. 6a: authors must clarify why the sensor works the best at pH 5.5. The description on high current at pH 5.5 is not sufficient to understand the phenomena between the WE materials and electrolyte.

Ans: Thanks for reviewer’s comment. The sensor works the best at pH 5.5 because of its stability. It undergoes denaturation in strongly acidic or alkaline solution, thus pH = 5.5 was suitable for CA125 immunosensor [Kalyani et al.2021].

  1. I could not understand the justification provided for the Fig. 6b. If authors were to select a constant current why don’t they select 15 µg/mL which is more economical compare to the 20 µg/mL.

Ans: Thanks for reviewer’s comment. It is a practise to select a standard system that is more reliable. Even though the current value of 15 µg/mL is similar to the 20 µg/mL, the later one was chosen for further reliable studies.

  1. For the calibration in Fig. 7 why authors chose Integral area as y axis instead of common practise Current?

Ans: Thanks for reviewer’s comment. It is usual to choose the current value. However, in the present study the integral area was chosen. The sensor studies notice changes in current along with curve widths. In order to calculate the correct current values, the integral area was chosen instead of current values in y-axis.

Reviewer 2 Report

The manuscript entitled “Improving the oxygen-containing functional groups of multi-walled carbon nanotubes to enhance the sensitivity of label-free electrochemical immunosensor for the detection of CA125” discusses about four different types of CBN electrochemical behavior against CA125. In short, the manuscript would have been clean and neat if the authors just focus on MWCNT. The investigation of other CBN derivatives makes the work quite extensive. However, materials characterizations and various electrochemical analyses provides significant input to the existing body of knowledge. So I have given some recommendation to improvise the manuscript before it can be considered for publication.

Title:

1.      Based on the manuscript – the study investigates on various types of CBNs. Therefore, it’s inappropriate to state one of the CBN-MWCNT’s name only in the title. It’s recommended to use general term – CBN. Suggested title: ” Improving the oxygen-containing functional groups of carbon-based nanomaterials to enhance the sensitivity of label-free electrochemical immunosensor for the detection of CA125”

Abstract:

1.      Page 1, Line 19-22: The labelling of modified electrode is too long. Be sure to include the significant materials only as your WE. In addition, I don’t find much info about mentioned materials in the Introduction section.

Introduction:

1.      Authors have used 4 different CBN – MWCNT, VGCFs, graphite KS4, and carbon black super P (SP). The Introduction section must contain related works to KS4 and SP. I can only find many literatures were based on MWCNT and only one work by Karuppiah et al on VGCFs. More LR must be added on KS4 and SP.

Methods:

1.      Page 4, Line 178-179. It was little confusing when authors mentioned about 3 electrode system then, all sampled were deposited on screen printed electrodes. The SPEs are for morphology purposes only. So I recommend to shift this method to the end of Section 2.2 where author explained about FESEM.

2.      Page 5, Line 207: What is rxn?

3.      Section 2.6 – many grammatical errors were spotted. RT is not a scientific style of writing. Use specific standard room temperature i.e. 25°C.

Results and Discussion:

1.      The resolution of inset of Fig 1b must be enhanced. Fig 1e does not reflect the MWCNT anymore. Kindly replace the Fig 1b where MWCNTs are visible within the scattered AuNPs.

2.      Page 7, Line 292 – authors stated the membranes. Why it was suddenly labelled as membranes?

3.      Author declared that, KS4 showed minor defect. However, according to Table S1 – I believe ID/IG increased from 0.240 to 0.472 which is a great enhancement. Kindly clarify.

4.      Page 9, Line 337: How do you arrange MWCNTs>SP>KS4>VGCFs in this order?

5.      The resolution of Fig. 4 must be enhanced.

6.      Section 3.5: First of all, authors mentioned about 7 parameters were optimized. However the results of 5 parameters only shown in the manuscript. The parameters such as Ab incubation time, Ab-Ag incubation time, Ag concentration missing in Fig. 6.

7.      Section 3.5, Line 451: “The maximum concentration for immobilization….” Since authors investigated many parameters, they must assure which parameter are they discussing when moving from one to another. Kindly rewrite the sentences to depict a clearer picture of these parameters.

8.      Fig. 6a: authors must clarify why the sensor works the best at pH 5.5. The description on high current at pH 5.5 is not sufficient to understand the phenomena between the WE materials and electrolyte.

9.      I could not understand the justification provided for the Fig. 6b. If authors were to select a constant current why don’t they select 15 µg/mL which is more economical compare to the 20 µg/mL.

10.   For the calibration in Fig. 7 why authors chose Integral area as y axis instead of common practise Current?

Author Response

(The authors gave the same response as above.)

Round 2

Reviewer 1 Report

Accept in present form.

Reviewer 2 Report

Authors have addressed all my comments and I agree for the publication of the manuscript in the present form.